# Learning to Branch with Tree MDPs

**Lara Scavuzzo**
Delft University of Technology
`l.v.scavuzzomontana@tudelft.nl`

**Feng Yang Chen**
Polytechnique Montréal
`feng-yang.chen@polymtl.ca`

**Didier Chételat**
Polytechnique Montréal
`didier.chetelat@polymtl.ca`

**Maxime Gasse**
Mila, Polytechnique Montréal
`maxime.gasse@polymtl.ca`

**Andrea Lodi**
Jacobs Technion-Cornell Institute
Cornell Tech and Technion - IIT
`andrea.lodi@cornell.edu`

**Neil Yorke-Smith**
Delft University of Technology
`n.yorke-smith@tudelft.nl`

**Karen Aardal**
Delft University of Technology
`k.i.aardal@tudelft.nl`

## Abstract

State-of-the-art Mixed Integer Linear Program (MILP) solvers combine systematic tree search with a plethora of hard-coded heuristics, such as the branching rule. The idea of learning branching rules from data has received increasing attention recently, and promising results have been obtained by learning fast approximations of the *strong branching* expert. In this work, we instead propose to learn branching rules from scratch via Reinforcement Learning (RL). We revisit the work of Etheve et al. [11] and propose tree Markov Decision Processes, or *tree MDPs*, a generalization of temporal MDPs that provides a more suitable framework for learning to branch. We derive a tree policy gradient theorem, which exhibits a better credit assignment compared to its temporal counterpart. We demonstrate through computational experiments that tree MDPs improve the learning convergence, and offer a promising framework for tackling the learning-to-branch problem in MILPs.

## 1 Introduction

Mixed Integer Linear Programs (MILPs) offer a powerful tool for modeling combinatorial optimization problems, and are used in many real-world applications [30]. The method of choice for solving MILPs to global optimality is the Branch-and-Bound (B&B) algorithm, which follows a divide-and-conquer strategy. A critical component of the algorithm is the *branching* rule, which is used to recursively partition the MILP search space using a space search tree. While there is little understanding of optimal branching decisions in general settings [25], the choice of the branching rule has a decisive impact on solving performance in practice [3]. In modern solvers, state-of-the-art performance is obtained using hard-coded branching rules that rely on heuristics designed by domain experts to perform well on a representative set of test instances [18].

In recent years, increasing attention has been given to approaches that use Machine Learning (ML) to obtain good branching rules and improve upon expert-crafted heuristics [5]. Such a statistical

36th Conference on Neural Information Processing Systems (NeurIPS 2022).

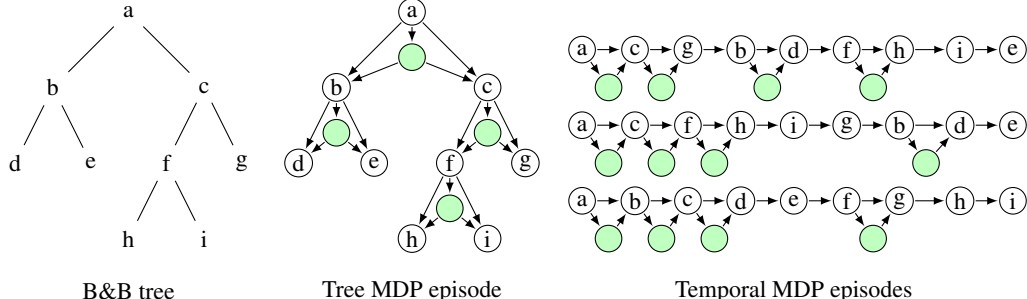

Figure 1: B&B process as a tree MDP episode vs. a temporal MDP episode. White nodes denote states, green nodes denote actions. In the tree MDP framework, the branching decision for splitting a node $f$ is credited two rewards, $(r_h, r_i)$. In the temporal MDP framework, the same branching decision is credited with additional rewards which depend on the temporal order in which B&B nodes are processed, $(r_h, r_i, r_e)$, $(r_h, r_i, r_g, r_b, r_d, r_e)$, or $(r_g, r_h, r_i)$.

approach is particularly well-suited in situations where similar problems are solved repeatedly, which is a common industrial scenario. In such cases, a generic and problem-agnostic branching rule might be suboptimal. Due to the sequential nature of branching, a natural paradigm to formulate the learning problem is Reinforcement Learning (RL), which allows to directly search for the optimal branching rule with respect to a metric of interest, such as average solving time, or final B&B tree size.

Etheve et al. [11] show that, when using depth-first-search (DFS) as the tree exploration (a.k.a. node selection) policy in B&B, minimizing each subtree size is equivalent to minimizing the global tree size. Using this result, they propose an efficient value-based RL procedure for learning to branch. In this paper we build upon the work of Etheve et al. [11], and we propose a generalization of the Markov Decision Process (MDP) paradigm that we call *tree MDP* (illustrated in Figure 1, formally discussed in Section 4). This general formulation captures the key insight of their approach, but opens the door for more general B&B settings, RL algorithms and reward functions.

In particular, we show that DFS is just one way to enforce the "tree Markov" property, and we propose an alternative, more practical solution that simply relies on providing the optimal objective limit to the solver during training. Our contribution is three-fold:

1. We introduce the concepts of a tree MDP and of the tree Markov property, and we derive a policy gradient theorem for tree MDPs.

2. We propose an alternative way to enforce the tree Markov property in learning to branch by imposing an optimal objective limit, which is less computationally demanding than DFS [11].

3. We show that adopting the tree MDP paradigm improves both the convergence speed of RL and the quality of the resulting branching rules, compared the regular MDP paradigm.

The remainder of this paper is organized as follows. We formally introduce MILPs, B&B and MDPs in Section 2. In Section 3 we motivate the need for moving beyond imitation learning in learning to branch, and discuss the challenges it raises. In Section 4 we introduce the concepts of tree MDPs and the tree Markov property, we derive a policy gradient theorem for tree MDPs, and we present a convenient solution to enforce the tree Markov property in B&B without DFS [11]. In Section 5 we conduct, for the first time, a thorough computational evaluation of RL for learning branching rules, on five synthetic MILP benchmarks with two difficulty levels. Finally, we discuss the significance of our results and future directions in Section 6.

## 2 Background

In this section, we describe the Branch-and-Bound (B&B) algorithm, and we show how the branching problem naturally formulates as a (temporal) Markov Decision Process.

## 2.1 The B&B algorithm

Consider a Mixed Integer Linear Program instance as an optimization problem of the form

$$\min_{x \in \mathbb{R}^n} \{c^T x : Ax \leq b, \, l \leq x \leq u, \, x_i \in \mathbb{Z} \;\; \forall i \in \mathcal{I}\}, \tag{1}$$

where $c \in \mathbb{R}^n$ is the objective coefficient vector, $A \in \mathbb{R}^{m \times n}$ is the constraint matrix, $b \in \mathbb{R}^m$ is the constraint right-hand-side, $l, u \in \mathbb{R}^n$ are the lower and upper bound vectors, respectively, and $\mathcal{I} \subseteq \{1, 2, ..., n\}$ is the subset of variables constrained to take integer values. Relaxing the integrality constraints, $x_i \in \mathbb{Z} \; \forall i \in \mathcal{I}$, yields a Linear Program (LP) that can be solved efficiently in practice, for example by using the Simplex algorithm. Solving such a relaxation yields an LP solution $\hat{x}^\star$, and provides a lower bound $c^T \hat{x}^\star$ to the original problem (1). If by chance $\hat{x}^\star$ satisfies the integrality constraints, $\hat{x}_i^\star \in \mathbb{Z} \; \forall i \in \mathcal{I}$, then it is also an optimal solution to (1). If, on the contrary, there is at least one variable $x_i, i \in \mathcal{I}$ such that $\hat{x}_i^\star$ is not an integer, then one can split the feasible region into two sub-problems, by imposing

$$x_i \leq \lfloor \hat{x}_i^\star \rfloor \qquad \text{or} \qquad x_i \geq \lceil \hat{x}_i^\star \rceil . \tag{2}$$

This partitioning step is known as branching. In its most basic form, the vanilla B&B algorithm recursively applies branching, thereby building a search space tree where each node has an associated local sub-MILP, with a local LP solution and associated local lower bound.[1] If the local LP solution satisfies the MILP integrality constraints, then it is termed a *feasible solution*, and it also provides an upper bound to (1). At any given time of the B&B process, a Global Lower Bound (GLB) to the original MILP can be obtained by taking the lowest of local lower bounds of the leaf nodes of the tree. Similarly, a Global Upper Bound (GUB) can be obtained by taking the lowest of the upper bounds so far.[2] The B&B process terminates when no more B&B leaf node can be partitioned with (2), i.e., all leaf nodes satisfy one of these conditions: the local LP has no solution (infeasible); the local lower bound is above the GUB (pruned); or the local LP solution satisfies the MILP integrality constraints (integer feasible). At termination, we have $GLB = GUB$, and the original MILP (1) is solved.

The branching problem, a.k.a. variable selection problem, is the task of choosing, at every B&B iteration, the variable $x_i$ that will be used to generate a partition of the form (2). While there is no universal metric to measure the quality of a branching rule, a common performance measure is the final size of the B&B tree (the smaller the better) [1].

## 2.2 Temporal MDPs

In the following, upper-case letters in italics denote random variables (e.g. $S, A$), while their lower-case counterparts denote their value (e.g. $s, a$) and their calligraphic counterparts their domain (e.g., $s \in \mathcal{S}, a \in \mathcal{A}$). We consider episodic Markov Decision Processes (MDPs) of the form $M = (\mathcal{S}, \mathcal{A}, p_{init}, p_{trans}, r)$, with states $s \in \mathcal{S}$, actions $a \in \mathcal{A}$, initial state distribution $p_{init}(s_0)$, state transition distribution $p_{trans}(s_{t+1}|s_t, a_t)$, and reward function $r : \mathcal{S} \to \mathbb{R}$. For simplicity, we assume finite episodes of length $T$, that is, $\tau = (s_0, a_0, \ldots, s_T), |\tau| = T$.[3] Together with a control mechanism represented by a stochastic policy $\pi(a_t|s_t)$, the MDP defines a probability distribution over trajectories, namely

$$p_\pi(\tau) = p_{init}(s_0) \prod_{t=0}^{|\tau|-1} \pi(a_t|s_t) p_{trans}(s_{t+1}|s_t, a_t).$$

The MDP control problem is to find a policy that maximizes the expected cumulative reward, $\pi^\star \in \arg\max_\pi V^\pi$, with

$$V^\pi := \mathop{\mathbb{E}}_{\tau \sim p_\pi} \left[ \sum_{t=0}^{|\tau|} r(s_t) \right] . \tag{3}$$

A key property of MDPs is the temporal Markov property $S_{t+1} \perp\!\!\!\perp S_{<t} \mid S_t, A_t, \forall t$, which guarantees that the future only depends upon the current state and action. One consequence of this property is

---

[1] The local lower bound is $\infty$ if the LP is infeasible.

[2] The local upper bound is $\infty$ if the LP is infeasible, or if its solution is not MILP feasible.

[3] Equivalently, we can assume all trajectories will reach an absorbing, zero-reward state $s_{null}$ in a finite number of steps, that is, $p_{trans}(s_{t+1}|s_t = s_{null}, a_t) = \delta_{s_{null}}(s_{t+1})$, $r(s_{null}) = 0$, and $\exists T$ s.t. $p_\pi(s_T = s_{null}) = 1, \forall \pi$.

the temporal policy gradient theorem [34], which forms the basis of policy gradient methods

$$\nabla_\pi V^\pi = \mathop{\mathbb{E}}_{\tau \sim p_\pi} \left[ \sum_{t=0}^{|\tau|-1} \nabla_\pi \log \pi(a_t|s_t) \sum_{t'=t+1}^{|\tau|} r(s_{t'}) \right]. \tag{4}$$

## 2.3 The branching temporal MDP

Let us now consider the problem of learning a branching policy in a B&B solver. As noted by [21, 17], branching decisions are made sequentially, thus the problem can naturally be regarded as a temporal MDP. In this form, the states $s_t$ consist of the entire state of the B&B process (that is, the solver) at time $t$, which includes the whole tree structure, all sub-MILPs and LP solutions, and all upper and lower bounds. The actions $a_t$ are the branching decisions, and the reward is chosen so that the return matches an objective function of interest (e.g., the final B&B tree size).

While appealing by its simplicity, we argue that this formulation is not practical for two reasons. First, even in the simplest textbook implementation of B&B, the MDP states are complex objects of growing size, which are impractical to handle. Second, episode length in branching can grow extremely large, in the worst case exponentially with the size of the problem. This exacerbates the so-called *credit assignment problem*, i.e., the problem of determining which actions should be given credit for a certain outcome (see Section 3.2 for a more detailed discussion). In Section 4, we will we will show how the tree MDP formulation addresses both those challenges.

# 3 Approaches to learning to branch

Approaches that frame branching as a learning problem have gained significant attention recently. In this section, we review some common trends in the field and identify some key challenges.

## 3.1 Learning to branch with imitation learning

In recent years, it has been demonstrated that efficient branching rules can be obtained by learning fast approximations of *strong branching*, a globally effective but computationally expensive rule [23, 26, 20, 17, 19, 36, 29]. This approach can be framed as imitation learning, a well-known method for learning in MDPs when an expert is available. While this approach can lead to improvements over state-of-the-art branching rules on various benchmarks, it also has several drawbacks.

First, strong branching implementations are known to trigger side effects (such as early detection of an infeasible child) that do not map well within the branching MDP framework [14]. This means that branching decisions collected from the strong branching expert might not line up with the environment of the learning agent, which might result in a performance gap between the learning agent and the expert, even if the agent manages to successfully reproduce the expert decisions.

Second, other than these side-effects, strong branching relies on dual bound improvements to make branching decisions. This can be ineffective in problems where the LP relaxation is not very informative or suffers from dual degeneracy, as pointed out in Gamrath et al. [16]. As an extreme case, Dey et al. [9] provide an example where a strong-branching based B&B tree can have exponentially more nodes than the tree obtained with a problem-specific rule.

Finally, regardless of the expert quality, obtaining strong branching samples can become prohibitively expensive for large instances. This means that non-trivial engineering solutions and scaling strategies are needed to allow training on larger problems, such as heavy computational parallelization [29].

## 3.2 Learning to branch with reinforcement learning

The reasons listed above suggest the need for an alternative approach to finding a good branching policy; an approach that does not rely on the strong branching rule. One could perform imitation learning on another expert, but no other plausible imitation target is known, and the same performance ceiling issue would remain. A natural alternative is reinforcement learning, an approach that aims to find an optimal policy in a Markov Decision Process, with respect to any desired objective function. But despite the theoretical appeal of RL for learning to branch, it also comes with its own challenges.

First, common evaluation metrics in B&B, such as solving time or final tree size, are inconvenient for RL. Both require to run episodes to completion, that is, to solve MILP instances to optimality, which leads to very long episodes even for moderately hard instances. Second, in contrast to many RL tasks studied in the literature, in B&B the worse the policy is, the longer are the episodes. These two factors combined give rise to the following problems:

1. Collecting training data is computationally expensive. In particular, training from scratch from a randomly initialized policy can be prohibitive for large MILPs.

2. Due to the length of the episodes, training signals are particularly sparse. This exacerbates the so-called *credit assignment problem* [27], i.e., the problem of determining which actions should be given credit for a certain outcome.

Two works have tried to tackle the branching problem with RL so far. Sun et al. [33] propose an approach based on evolution strategies, a variant of RL, combined with a novelty bonus based on discrepancies in B&B trees. They show improvements over an imitation learning approach Gasse et al. [17] on instances derived from a common backbone graph, but fail to improve on more heterogeneous ones. These results emphasize the difficulty of a direct application of RL on hard instance sets.

In parallel, Etheve et al. [11] recently made a contribution that directly addresses the credit assignment problem. They show that, when depth-first-search (DFS) is used as the node selection strategy, minimization of the total B&B tree size can be achieved by taking decisions that minimize subtree size at each node. Based on this result, they propose a Q-learning-type algorithm [28] where the learned Q-function approximates the local subtree size. They report improvements over a state-of-the-art branching rule on collections of small, fixed-size instances. However, they only evaluate the learned policy with DFS node selection, which matches their training setting, but is not a realistic B&B setting. In Section 4, we will show that the method proposed by Etheve et al. [11] can be interpreted as a specific instantiation of a more general *tree MDP* framework, which effectively simplifies the credit assignment problem in learning to branch (point 2 above). Furthermore, we propose an alternative condition to the one of Etheve et al. [11] to ensure a tree MDP setting, which results in shorter episodes, hence reducing the cost of data collection (point 1 above).

## 4 Branching as a tree MDP

We now detail our tree MDP framework, and show how the branching problem can be cast as a tree MDP control problem, under some conditions. Proofs as well as a side-by-side comparison of temporal and tree MDPs are deferred to the Supplementary Material.

### 4.1 Tree MDPs

We define tree MDPs as augmented Markov Decision Processes $tM = (\mathcal{S}, \mathcal{A}, p_{init}, p_{ch}^-, p_{ch}^+, r, l)$, with states $s \in \mathcal{S}$, actions $a \in \mathcal{A}$, initial state distribution $p_{init}(s_0)$, respectively left and right child transition distributions $p_{ch}^-(s_{ch_i^-}|s_i, a_i)$ and $p_{ch}^+(s_{ch_i^+}|s_i, a_i)$[4], reward function $r : \mathcal{S} \to \mathbb{R}$ and leaf indicator $l : \mathcal{S} \to \{0, 1\}$. The central concept behind tree MDPs is that each non-leaf state $s_i$ (i.e., such that $l(s_i) = 0$), together with an action $a_i$, produces two new states $s_{ch_i^-}$ (its left child) and $s_{ch_i^+}$ (its right child). As a result, the tree MDP generative process results in episodes $\tau$ that follow a tree structure (see Figure 1), where leaf states (i.e., such that $l(s_i) = 1$) are the leaf nodes of the tree, below which no action can be taken and no children state will be created. For simplicity, just like in Section 2.2, we assume the tree-like trajectories have some finite size that we denote $T = |\tau|$.

A tree MDP episode $\tau$ consists of a binary[5] tree with nodes $\mathcal{N} = \{0, \dots, |\tau|\}$ and leaf nodes $\mathcal{L} = \{i | i \in \mathcal{N}, l(s_i) = 1\}$, which embeds a state $s_i$ at every node $i \in \mathcal{N}$ and an action $a_i$ at every non-leaf node $i \in \mathcal{N} \setminus \mathcal{L}$. For convenience, in the following we will use $pa_i$, $ch_i^-$ and $ch_i^+$ to denote the nodes that are respectively parent, left child and right child of a node $i$ if any, as well as $d_i$ and $nd_i$ to denote respectively the set of all descendants and non-descendants of a node $i$ in the tree. Together

---

[4]In the case of branching, these transitions are deterministic.
[5]The concept can easily be extended to non-binary trees.

with a control mechanism $\pi(a_t|s_t)$, a tree MDP defines a probability distribution over trajectories,

$$p_\pi(\tau) = p_{init}(s_0) \prod_{i \in \mathcal{N} \setminus \mathcal{L}} \pi(a_i|s_i) p_{ch}^-(s_{ch_i^-}|s_i, a_i) p_{ch}^+(s_{ch_i^+}|s_i, a_i).$$

As in temporal MDPs, the tree MDP control problem is to find a policy that maximizes the expected cumulative reward, as defined by (3). Due to their specific generative process, a key characteristic of tree MDPs is the *tree Markov property* $S_{ch_i^-}, S_{ch_i^+} \perp\!\!\!\perp S_{nd_i} \mid S_i, A_i, \forall i$, which guarantees, similarly to the temporal Markov property, that each subtree only depends upon the immediate state and action. This again simplifies the credit assignment problem in RL and results in an efficient *tree policy gradient* formulation.

**Proposition 4.1.** *For any tree MDP tM, the policy gradient can be expressed as*

$$\nabla_\pi V^\pi = \mathop{\mathbb{E}}_{\tau \sim p_\pi} \left[ \sum_{i \in \mathcal{N} \setminus \mathcal{L}} \nabla_\pi \log \pi(a_t|s_t) \sum_{j \in d_i} r(s_j) \right]. \tag{5}$$

## 4.2 The branching tree MDP

We now show how and under which conditions the vanilla B&B algorithm can be formulated as a tree MDP. We consider episodes $\tau$ that follow exactly the B&B tree structure. Each node $i$ in the tree embeds a state $s_i = (MILP_i, GUB_i)$, where $MILP_i$ is the local sub-MILP of the node, and $GUB_i$ is the global upper bound at the time the node is processed[6]. Each non-leaf node also embeds an action $a_i = (j, x_j^\star)$, where $j$ is the index of the branching variable chosen by B&B, and $x_j^\star$ is the value used to branch ($x_j \leq \lfloor x_j^\star \rfloor \vee x_j \geq \lceil x_j^\star \rceil$). Note that such states and actions, embedded in the B&B tree, carry enough information to unroll a vanilla B&B algorithm, as described in Section 2.1. We now need to make two additional assumptions in order to formulate branching as a tree MDP.

### 4.2.1 B&B tree transitions

First, and this is our main requirement, B&B state transitions must decompose into $p_{ch}^-$ and $p_{ch}^+$.

**Assumption 4.2.** For every non-leaf node $i$, the global upper bounds $GUB_{ch_i^-}$ and $GUB_{ch_i^+}$ (reached by B&B when the left and right child is processed, respectively) can be derived solely from the current state and action, $(s_i, a_i)$.

**Proposition 4.3.** *A vanilla B&B algorithm that satisfies Assumption 4.2 forms a tree MDP.*

Assumption 4.2 is not always satisfied, as the following counterexample shows.
*Counter example.* Consider the root problem $MILP_0 = \min x \ s.t. \ x \geq 0.6, \ x \in \mathbb{Z}$, with upper bound $GUB_0 = \infty$. The root LP solution is $\hat{x}^\star = 0.6$, and the two sub-problems $MILP_{ch_i^-}$ and $MILP_{ch_i^+}$ follow from the (only) branching decision $x \leq 0 \vee x \geq 1$. Now, the two global upper bound $GUB_{ch_i^-}$ and $GUB_{ch_i^+}$ depend on whether the feasible solution $x = 1$ has been found in the past, which in turn depends on the node processing order. Going left first ($-$) will yield $(GUB_{ch_i^-}, GUB_{ch_i^+}) = (\infty, \infty)$, while going right first ($+$) will yield $(GUB_{ch_i^-}, GUB_{ch_i^+}) = (1, \infty)$.

We now provide two conditions under which Assumption 4.2 is true.

**Proposition 4.4.** *In Optimal Objective Limit B&B (ObjLim B&B), that is, when the optimal solution value of the MILP is known at the start of the algorithm ($GUB_0 = GUB^\star$), Assumption 4.2 holds.*

**Proposition 4.5.** *In Depth-First-Search B&B (DFS B&B), that is, when nodes are processed depth-first and left-first by the algorithm, Assumption 4.2 holds.*

Propositions 4.4 and 4.5 provide two viable options for turning vanilla B&B into a tree MDP, where $p_{ch}^-$ and $p_{ch}^+$ are deterministic functions. The first variant, *ObjLim*, requires MILP instances used for training to be solved to optimality once, in order to collect their optimal objective value. The second variant, *DFS*, corresponds to the setting in [11]. In this variant there is no need to pre-solve training instances to optimality, however it is expected that the collected episodes might be longer than with a standard node selection rule, which might result in slower training.

---

[6]Recall that in B&B, the GUB corresponds to the value of the best feasible solution found so far, or $\infty$ when no solution has been found yet.

### 4.2.2 B&B tree reward

Last, for branching to formulate as a control problem in a tree MDP, the objective must be compatible.

**Assumption 4.6.** The branching objective can be decomposed over the nodes of the B&B tree, with a state-based reward function $r : \mathcal{S} \rightarrow \mathbb{R}$.

Interestingly, a natural objective for branching is the final *B&B tree size*, which expresses naturally as $r(s_i) = -1$. Thus, it is trivially compatible with Assumption 4.6. We will consider this reward in our experiments. Another common objective is the total solving time, which can also be expressed as a state-based reward $r : \mathcal{S} \rightarrow \mathbb{R}$ under mild assumptions. Indeed, it suffices to considers that solving LP relaxations and making branching decisions at each node are the main contributing factors in the total running time, while other algorithmic components have a negligible cost. In vanilla B&B, both these components only depend on the local state $s_i = (MILP_i, GUB_i)$ of each node.

### 4.3 Efficiency of tree MDP

Tree MDPs, when applicable, provide a convenient alternative to temporal MDPs for tackling the branching problem. First, the tree Markov property implies that branching policies in tree branching MDPs will not benefit from any information other than the local state $s_i = (MILP_i, GUB_i)$ to make optimal decisions, similarly to how control policies in temporal MDP can ignore past states and rely only on the immediate state $s_t$. Second, the credit assignment problem in the branching tree MDP is more efficient than in the equivalent temporal MDP. This is showcased in Figure 1, and stems from the fact that in a tree MDP all the descendants of a node $i$ are necessarily processed after that node temporally. As a consequence, the rewards credited to an action in the tree policy gradient (5), $\sum_{j \in d_i} r(s_j)$, are necessarily a subset of the rewards credited to the same action in the temporal policy gradient (4), $\sum_{t' > t} r(s_{t'})$. Thus, it can be expected intuitively that learning branching policies within the tree MDP framework will be easier, and more sample-efficient than learning within the temporal MDP framework. We will validate this hypothesis experimentally in Section 5.

### 4.4 Theoretical limitations

Our proposed B&B variants, *ObjLim* and *DFS*, allow for a nice formulation of the branching problem as a tree MDP, which we argue is key to unlocking a more practical and sample-efficient learning of branching policies. However, usually the end goal is to learn a branching policy that performs well in realistic B&B settings, and the fact that a branching policy performs well in one of those variants does not guarantee that it will perform well in the vanilla setting also. This discrepancy between the training environment and the evaluation environment is a recurring problem in RL, and is more generally referred to as the transfer learning problem. While there exist solutions to mitigate this problem, in this paper we leave the question aside and simply assume that the transfer problem is negligible. We thus directly report the performance obtained from each training setting in the realistic evaluation setting, a default B&B solver.

### 4.5 Connections with hierarchical RL

The tree MDP formulation has connections with hierarchical RL (HRL), a paradigm that aims at decomposing the learning task into a set of simpler tasks that can be solved recursively, independently of the parent task. The most related HRL approach is perhaps MAXQ [10], which decomposes the value function of an MDP recursively using a finite set of smaller constituent MDPs, each with its own action set and reward function. For example, delivering a package from a point A to a point B decomposes into: moving to A, picking up package, moving to B, dropping package. While both tree MDP and MAXQ exploit a recursive tree decomposition in order to simplify the credit assignment problem, the two frameworks also differ on several points. First, in MAXQ the hierarchical sub-task structure must be known a priori for each new task, and results in a fixed, limited tree depth, while in tree MDPs the decomposition holds by construction and can result in virtually infinite depths. Second, in MAXQ each sub-task results in a different MDP, while in tree MDPs all sub-tasks are the same. Lastly, in MAXQ the recursive decomposition must follow a temporal abstraction, where each episode is processed according to a depth-first traversal of the tree. In tree MDPs the decomposition is not tied to the temporal processing order of the episode, except for the requirement that a parent must be processed before its children. Thus, any tree traversal order is allowed (see Figure 1).

---

**Algorithm 1** REINFORCE training loop

---

1: **Input:** training set of MILP instances and their pre-computed optimal solution $\mathcal{D}$, maximum
   number of epochs $K$, time limit $\zeta$, entropy bonus $\lambda$, learning rate $\alpha$, sample rate $\beta$.
2: Initialize policy $\pi_\theta$ with random parameters $\theta$.
3: **for** *epoch* from 1 to $K$ **do**
4:     **if** elapsed time $> \zeta$ **then** break
5:     Sample 10 MILP instances from $\mathcal{D}$
6:     **for** each sampled instance **do**
7:       Collect one episode $\tau$ by running B&B to optimality
8:       Extract randomly $\beta \times |\tau|$ state, action, return tuples $(s, a, G)$ from $\tau$ (with $G$ the local
   subtree size for tree MDPs, and the remaining episode size for MDPs)[8]
9:     **end for**
10:    $n \leftarrow$ number of collected tuples, $L \leftarrow 0$
11:    **for** each collected tuple $(s, a, G)$ **do**
12:      $L \leftarrow L - G\frac{1}{n}\log \pi_\theta(a|s)$ # policy gradient cost
13:      $L \leftarrow L - \lambda\frac{1}{n}H(\pi_\theta(\cdot|s))$ # entropy bonus
14:    **end for**
15:    $\theta \leftarrow \theta - \alpha\nabla_\theta L$
16: **end for**
17: **return** $\pi_\theta$

---

## 5 Experiments

We now compare the performance of four machine learning approaches: the imitation learning
method of Gasse et al. [17], and three different RL methods. We also compare against SCIP's default
rule (for a description of this rule we refer to the Supplementary Material A.4). Code for reproducing
all experiments is available online [7].

### 5.1 Setup

**Benchmarks** Similarly to Gasse et al. [17], we train and evaluate each method on five NP-hard
problem benchmarks, which consist of synthetic combinatorial auctions, set covering, maximum
independent set, capacitated facility location and multiple knapsack instances. For each benchmark
we generate a training set of 10,000 instances, along with a small set of 20 validation instances for
tracking the RL performance during training. For the final evaluation, we further generate a test set
of 40 instances, the same size as the training ones, and also a transfer set of 40 instances, larger and
more challenging than the training ones. More information about benchmarks and instance sizes can
be found in the Supplementary Material (A.5).

**Training** We use the Graph Neural Network (GNN) from Gasse et al. [17] to learn branching
policies, with the same features and architecture. This state representation has been shown to provide
an efficient encoding of the local MILP together with some global features. Notice that this makes
the MDP formulation into a POMDP, given that we do not encode the complete search tree (see
Section 2.3). We compare four training methods: imitation learning from strong branching (IL); RL
using temporal policy gradients (MDP); RL using tree policy gradients with DFS as a node selection
strategy (tMDP+DFS), which enforces the tree Markov property due to Proposition 4.5; and RL
using tree policy gradients with the optimal objective value set as an objective limit (tMDP+ObjLim),
which corresponds to Propositions 4.4. Other than that, we use default solver parameters, except for
restarts and cutting planes after the root node which are deactivated. We use a plain REINFORCE
[35] with entropy bonus as our RL algorithm, for simplicity. Our training procedure is summarized
in Algorithm 1. We set a maximum of 15,000 epochs and a time limit of six days for training.
Our implementation uses PyTorch [31] together with PyTorch Geometric [12], and Ecole [32] for
interfacing to the solver SCIP [15]. All experiments are run on compute nodes equipped with a GPU.

---

[7]`https://github.com/lascavana/rl2branch`
[8]Notice that for tree MDPs the computation of the return for each node can be computed efficiently with a
bottom-up traversal that runs in $O(n)$.

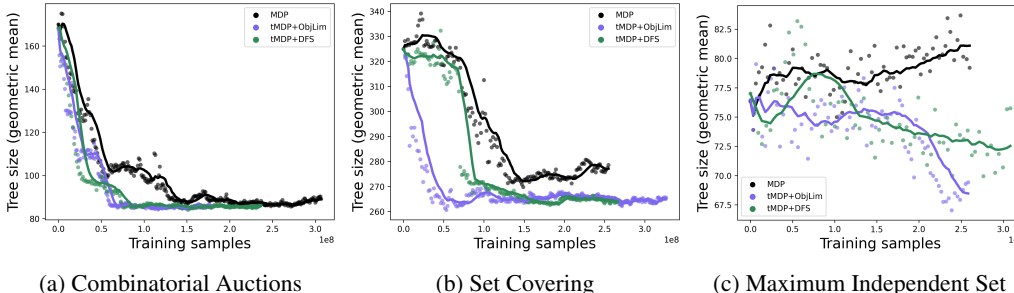

|  (a) Combinatorial Auctions | (b) Set Covering | (c) Maximum Independent Set |

Figure 2: Training curves for REINFORCE with temporal policy gradients (MDP), tree policy gradients with objective limit (tMDP+ObjLim) and DFS node selection (tMDP+DFS). We report the final B&B tree size on the validation set (geometric mean over 20 instances × 5 seeds, the lower the better), versus the number of processed training samples on the x-axis. Solid lines show the moving average. Training curves for the remaining benchmarks can be found in the Supplementary Material (A.3).

**Evaluation**    For each branching rule evaluated, we solve every validation, test or transfer instance 5 times with a different random seed. We use default solver parameters, except for restarts and cutting planes after the root node which are deactivated (same as during training), and a time limit of 1 hour for each solving run. For tMDP+DFS and tMDP+ObjLim, the specific settings used during training (DFS node selection and optimal objective limit, respectively) are not used any more, thus providing a realistic evaluation setting. We report the geometric mean of the final B&B tree size as our metric of interest, as is common practice in the MILP literature [3]. We pair this with the average per-instance standard deviation (in percentage). We only consider solving runs that finished successfully for all methods, as in [17]. Extended results including solving times are provided in the Supplementary Material (A.3).

## 5.2   Results

Figure 2 showcases the convergence of our three RL paradigms MDP, tMDP+ObjLim and tMDP+DFS during training, in terms of the final B&B tree size on the validation set (the lower the better). In order to better highlight the sample efficiency of each method, we report on the x-axis the cumulative number of collected training samples, which correlates with the length of the episodes collected during training. This provides a hardware-independent proxy for training time. As can be seen, the tree MDP paradigm clearly improves the convergence speed on these three benchmarks, with a clear domination of tMDP+ObjLim on one benchmark (Set Covering). Training curves for the remaining benchmarks are available in the Supplementary Material (A.3).

Table 1 reports the final performance of the branching rules obtained with each method, on both a held-out test set (same instance difficulty as training) and a transfer set (larger, more difficult instances than training). Despite a mismatch between the training and evaluation environments, which is required to enforce the tree Markov property, the tree MDP paradigm consistently produces equal or better branching rules than the temporal MDP paradigm on all 5 benchmarks.

On one benchmark, Multiple Knapsack, the branching rules learned by RL outperform both SCIP's default branching rule and the strong branching imitation (IL) approach. The likely reason is that the MILP formulation of Multiple Knapsack provides a very poor linear relaxation, which often results in no dual bound improvement after branching. This means that strong branching scores are in most cases not discriminative, which is problematic for rules that heavily rely on this criterion (see Section 3.1), such as SCIP's default or a policy that imitates strong branching. This situation makes a strong case for the potential of RL-based methods, which can adapt and devise alternative branching strategies.

On the remaining 4 benchmarks, however, RL methods perform worse than SCIP default or IL, despite being based on the same GNN architecture. This illustrates the difficulty of learning to branch via RL, even on small-scale problems, and the remaining room for improvement. Additional evaluation criteria (solving times and number of time limits) are available in the Supplementary Material (A.3).

Table 1: Solving performance of the different branching rules in terms of the final B&B tree size (lower is better). We evaluate each method on a test set with instances the same size as training, and a transfer set with larger instances. We report the geometric mean and standard deviation over 40 instances, solved 5 times with different random seeds, and we bold the best of the RL methods.

| Model | Comb. Auct. | Set Cover | Max.Ind.Set | Facility Loc. | Mult. Knap. |
|---|---|---|---|---|---|
| SCIP default | 7.3±39% | 10.7±24% | 19.3±52% | 203.6±63% | 267.8±96% |
| IL | 52.2±13% | 51.8±10% | 35.9±36% | 247.5±39% | 228.0±95% |
| RL (MDP) | 86.7±16% | 196.3±20% | 91.8±56% | 393.2±47% | 143.4±76% |
| RL (tMDP+DFS) | **86.1±17%** | **190.8±20%** | 89.8±51% | 360.4±46% | **135.8±75%** |
| RL (tMDP+ObjLim) | 87.0±18% | 193.5±23% | **85.4±53%** | **325.4±41%** | 142.4±78% |

Test

| Model | Comb. Auct. | Set Cover | Max.Ind.Set | Facility Loc. | Mult. Knap. |
|---|---|---|---|---|---|
| SCIP default | 733.9±26% | 61.4±19% | 2867.1±35% | 344.3±57% | 592.3±75% |
| IL | 805.1±9% | 145.0±6% | 1774.8±38% | 407.8±37% | 1066.1±101% |
| RL (MDP) | 1906.3±18% | 853.3±27% | 2768.5±76% | 679.4±52% | 518.4±79% |
| RL (tMDP+DFS) | **1804.6±17%** | **816.8±25%** | 2970.0±76% | 609.1±47% | 495.1±81% |
| RL (tMDP+ObjLim) | 1841.9±18% | 826.4±26% | **2763.6±74%** | **496.0±48%** | **425.3±64%** |

Transfer

# 6 Conclusions and Future Directions

This paper adds to a growing body of literature on using ML to assist decision-making in several key components of the B&B algorithm (see e.g. [5, 7, 22]). We contribute to the study of RL as a tool for learning to branch in MILP solvers. We present tree MDP, a variant of Markov Decision Processes, and show that under some conditions, the B&B branching process is tree-Markovian. We show that the approach of Etheve et al. [11] can be naturally cast as Q-learning for tree MDPs, and we propose an alternative, more computationally appealing way to enforce the tree Markov property in B&B, using optimal objective limits. Finally, we evaluate for the first time a variety of RL-based branching rules in a comprehensive computational study, and we show that tree MDPs improve the convergence speed of RL for branching, as well as the overall performance of the learnt branching rules.

These contributions bring us closer to learning efficient branching rules from scratch using RL, which could ultimately outperform existing branching heuristics built upon decades of expert knowledge and experiment. However, despite the convergence speed-up that our method provides, training without expert knowledge remains very computationally heavy and in general still results in worse performance than its supervised learning counterpart, which reveals a significant gap that must be closed. As future work, we would like to explore ideas to keep improving sample efficiency, and generalization across instance size. This is necessary for RL to scale to larger, non-homogeneous benchmarks, such as MIPLIB [18], which at the moment remain out-of-reach for RL.

Finally, although our concern in this paper is focused on improving variable selection for MILP, our tree MILP construction could be useful in other applications. Branch-and-bound is a type of divide-and-conquer algorithm, and we expect that, in general, this framework can be applied to any problem where one seeks to control such algorithms more efficiently. Examples would include controlling the order in which a rover explores rooms in a building, or selecting the pivots in a quicksort algorithm.

## Acknowledgements

The authors thanks the anonymous reviewers for their suggestions. The work of Karen Aardal, Lara Scavuzzo and Neil Yorke-Smith was partially supported by TAILOR, a project funded by EU Horizon 2020 research and innovation programme under grant number 952215, and by The Netherlands Organisation for Scientific Research (NWO), grant OCENW.GROOT.2019.015. The work of Feng Yang Chen, Didier Chételat, Maxime Gasse and Andrea Lodi was supported by the Canada Excellence Research Chairs program (CERC).

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
