# A  Supplementary Material

## A.1  Side-by-side comparison of MDP and tMDP

**Temporal MDP**

| | |
|---|---|
| A temporal MDP process: | $(\mathcal{S}, \mathcal{A}, p_{init}, p_{trans}, r)$ |
| Probability of a trajectory $\tau$: | $p_\pi(\tau) = p_{init}(s_0) \prod_{t=0}^{|\tau|-1} \pi(a_t|s_t) p_{trans}(s_{t+1}|s_t, a_t)$ |
| Markov property: | $S_{t+1} \perp\!\!\!\perp S_{<t} \mid S_t, A_t, \forall t$ |

**Tree MDP**

| | |
|---|---|
| A tree MDP process: | $(\mathcal{S}, \mathcal{A}, p_{init}, p_{ch}^-, p_{ch}^+, r, l)$ |
| Probability of a trajectory $\tau$: | $p_\pi(\tau) = p_{init}(s_0) \prod_{i \in \mathcal{N} \setminus \mathcal{L}} \pi(a_i|s_i) p_{ch}^-(s_{ch_i^-}|s_i, a_i) p_{ch}^+(s_{ch_i^+}|s_i, a_i).$ |
| Markov property: | $S_{ch_i^-}, S_{ch_i^+} \perp\!\!\!\perp S_{nd_i} \mid S_i, A_i, \forall i$ |

## A.2  Proofs

**Proposition 4.1.** *For any tree MDP tM, the policy gradient can be expressed as*

$$\nabla_\pi V^\pi = \mathop{\mathbb{E}}_{\tau \sim p_\pi} \left[ \sum_{i \in \mathcal{N} \setminus \mathcal{L}} \nabla_\pi \log \pi(a_t|s_t) \sum_{j \in d_i} r(s_j) \right]. \tag{5}$$

*Proof.* This proof draws closely to the proof of the temporal policy gradient theorem. First, let us re-write (3) as

$$V^\pi = \mathbb{E}_{s_0 \sim p_\pi} [V^\pi(s_0)],$$

where

$$V^\pi(s_i) := r(s_i) \qquad \text{if } l(s_i) = 1 \text{ (leaf node), and}$$

$$V^\pi(s_i) := r(s_i) + \mathbb{E}_{a_i, s_{ch_i^-}, s_{ch_i^+} \sim p_\pi} \left[ V^\pi(s_{ch_i^-}) + V^\pi(s_{ch_i^+}) \right] \qquad \text{if } l(s_i) = 0 \text{ (non-leaf node).}$$

The corresponding gradients when $l(s_i) = 1$ and $l(s_i) = 0$ are, respectively,

$$\nabla_\pi V^\pi(s_i) = 0, \text{ and}$$

$$\nabla_\pi V^\pi(s_i) = \mathbb{E}_{a_i, s_{ch_i^-}, s_{ch_i^+} \sim p_\pi} \left[ \frac{\nabla_\pi \pi(a_i|s_i)}{\pi(a_i|s_i)} \left( V^\pi(s_{ch_i^-}) + V^\pi(s_{ch_i^+}) \right) \nabla_\pi V^\pi(s_{ch_i^-}) + \nabla_\pi V^\pi(s_{ch_i^+}) \right].$$

Let us now write the gradient of $V^\pi$,

$$\nabla_\pi V^\pi = \mathbb{E}_{s_0 \sim p_{init}} [\nabla_\pi V^\pi(s_0)].$$

Either we have $l(s_0) = 1$ and thus $\nabla_\pi V^\pi = 0$, or we can expand $\nabla_\pi V^\pi(s_0)$ to obtain

$$\nabla_\pi V^\pi = \mathbb{E}_{s_0, a_0, s_{ch_0^-}, s_{ch_0^+} \sim p_\pi} \left[ \frac{\nabla_\pi \pi(a_0|s_0)}{\pi(a_0|s_0)} (V^\pi(s_{ch_0^-}) + V^\pi(s_{ch_0^+})) + \nabla_\pi V^\pi(s_{ch_0^-}) + \nabla_\pi V^\pi(s_{ch_0^+}) \right].$$

Then again, each of of the terms $\nabla_\pi V^\pi(s_{ch_0^-})$ and $\nabla_\pi V^\pi(s_{ch_0^+})$ can be replaced by 0 if the corresponding node is a leaf node, or can be expanded further in the same way if it is a non-leaf node. By

applying this rule recursively, we finally obtain

$$\nabla_\pi V^\pi = \mathbb{E}_{\tau \sim p_\pi} \left[ \sum_{i \in \mathcal{N} \backslash \mathcal{L}} \frac{\nabla_\pi \pi(a_i|s_i)}{\pi(a_i|s_i)} (V^\pi(s_{ch_i^-}) + V^\pi(s_{ch_i^+})) \right]$$

$$= \mathbb{E}_{\tau \sim p_\pi} \left[ \sum_{i \in \mathcal{N} \backslash \mathcal{L}} \frac{\nabla_\pi \pi(a_i|s_i)}{\pi(a_i|s_i)} \sum_{j \in d_i} r(s_j) \right]$$

$$= \mathbb{E}_{\tau \sim p_\pi} \left[ \sum_{i \in \mathcal{N} \backslash \mathcal{L}} \nabla_\pi \log \pi(a_i|s_i) \sum_{j \in d_i} r(s_j) \right].$$

$\square$

**Lemma A.1.** *In B&B, both children MILPs $MILP_{ch_i^-}$ and $MILP_{ch_i^+}$ can be derived from the local MILP $MILP_i$ and branching decision $a_i = (j, x_j^\star)$, with $j$ the index of a variable in $MILP_i$, and $x_j^\star$ the value to be used for branching.*

*Proof.* From the definition of B&B in Section 2, $MILP_{ch_i^-}$ (resp. $MILP_{ch_i^+}$) consist of $MILP_i$ augmented with the additional constraint $x_j \leq \lfloor x_j^\star \rfloor$ ( resp. $x_j \geq \lceil x_j^\star \rceil$). $\square$

**Proposition 4.3.** *A vanilla B&B algorithm that satisfies Assumption 4.2 forms a tree MDP.*

*Proof.* We shall now prove that, under Assumption 4.2, the B&B process can be formulated as a tree MDP $tM = (\mathcal{S}, \mathcal{A}, p_{init}, p_{ch}^-, p_{ch}^+, r, l)$, with states $s_i = (MILP_i, GUB_i)$ and actions $a_i = (j, x_j^\star)$. First, the algorithm starts at the root node with an initial MILP, $MILP_0$, and an initial global upper bound $GUB_0 = \infty$. Thus, the root state $s_0$ follows an arbitrary, user-defined MILP distribution $p_{init}(s_0)$, which is independent of the B&B algorithm. Second, Lemma A.1, together with Assumption 4.2, ensures the existence of (deterministic) distributions $p_{ch}^-(s_{ch_i^-}|s_i, a_i)$ and $p_{ch}^+(s_{ch_i^+}|s_i, a_i)$, from which the B&B children states $s_{ch_i^-}$ and $s_{ch_i^+}$ are generated. Third, the reward function $r(s_i)$ is not part of the B&B algorithm, and can be arbitrarily defined to match any (compatible) B&B objective. Last, the leaf node indicator $l(s_i)$ is exactly the vanilla B&B leaf node criterion, and is obtained by solving the LP relaxation of $MILP_i$ constrained with upper bound $GUB_i$, which results in either an infeasible LP (leaf node), a MILP-feasible LP solution (leaf node), or a MILP-infeasible LP solution (non-leaf node). This concludes the proof. $\square$

**Proposition 4.4.** *In Optimal Objective Limit B&B (ObjLim B&B), that is, when the optimal solution value of the MILP is known at the start of the algorithm ($GUB_0 = GUB^\star$), Assumption 4.2 holds.*

*Proof.* Because $GUB_0 = GUB^\star$, the initial global upper bound is equal to the optimal solution value to the original MILP. Then, B&B will never be able to find a feasible solution that tightens that bound, and we necessarily have $GUB_i = GUB_0, \forall i$. Hence $GUB_{ch_i^-} = GUB_{ch_i^+} = GUB_i$, and both $GUB_{ch_i^-}$ and $GUB_{ch_i^+}$ can be directly derived from $s_i$. This concludes the proof. $\square$

**Proposition 4.5.** *In Depth-First-Search B&B (DFS B&B), that is, when nodes are processed depth-first and left-first by the algorithm, Assumption 4.2 holds.*

*Proof.* First, it is trivial to show that $GUB_{ch_i^-}$ can be derived from $s_i$. Because node $i$ is not a leaf node, it has not resulted in an integral solution, and hence processing node $i$ does not change the GUB. And since $ch_i^-$ is processed directly after node $i$, we necessarily have $GUB_{ch_i^-} = GUB_i$. This, combined with Lemma A.1, shows that $s_{ch_i^-}$ can be inferred from $s_i$ and $a_i$. Second, we show how $GUB_{ch_i^+}$ can be derived from $s_i$ and $a_i$. Because node $ch_i^+$ is processed right after the whole subtree below $ch_i^-$ has been processed, $GUB_{ch_i^+}$ is necessarily the minimum of $GUB_i$ and the optimal solution value of $MILP_{ch_i^-}$. Now, because $s_{ch_i^-}$ can be inferred from $s_i$ and $a_i$, $MILP_{ch_i^-}$ can be recovered as well, and solved to obtain its optimal solution value. Therefore, $GUB_{ch_i^+}$ can be recovered from $s_i$ and $a_i$. This, together with $GUB_{ch_i^-} = GUB_i$, concludes the proof. $\square$

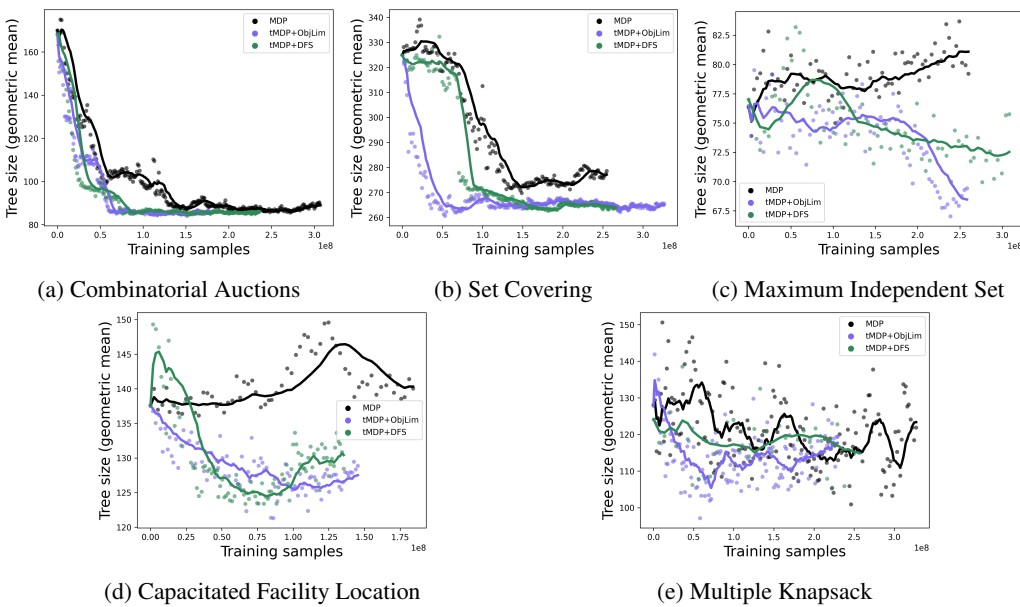

(a) Combinatorial Auctions      (b) Set Covering      (c) Maximum Independent Set

(d) Capacitated Facility Location      (e) Multiple Knapsack

Figure 3: All training curves. We report the final B&B tree size (geometric mean over 20 validation instances × 5 seeds, the lower the better). On the x-axis we report the number of processed training samples. Solid lines show the moving average. The compared methods are REINFORCE with temporal policy gradients (MDP), with tree policy gradients and objective limit (tMDP+ObjLim), and with tree policy gradients and depth-first search node selection (tMDP+DFS).

## A.3 Extended results

Here we provide all training curves (Figure 3) and the extended evaluation results (Table 2) with the geometric mean of the solving times in seconds (Time) and the geometric mean of the final B&B tree size (Nodes). The results are averaged over the solving runs that finished successfully for all methods. This is, if a solving run reached the time limit for any method, this is excluded from the average. Table 3 shows the number of solving runs that timed out per method.

## A.4 SCIP's default branching rule

SCIP assigns maximum priority by default to the hybrid branching rule [2]. This means that the choice of branching variable is based on a weighted sum of different criteria. The biggest weight is placed on the variable's pseudocosts. A variable's pseudocost is calculated as a function of the change in LP objective value we observe (on each of the branches) as a consequence of branching on that variable. This value can be explicitly calculated by tentatively branching on candidate variables (in which case the rule is called strong branching), or estimated based on past observed values. SCIP runs strong branching until it has stored a sufficient amount of observations for each variable, and then switches to the estimation strategy. Other than pseudocosts, SCIP also considers information about the implied reductions of other variables' domains and conflicts where the variable is involved, though with smaller importance.

It is important to consider that a call to strong branching can trigger a series of side-effects within the solver that are not accounted for in the node count. This was first observed by Gamrath and Schubert [14], who point out that this gives an unfair advantage to methods that use strong branching when comparing branching rules according to the final tree size.

## A.5 Instance collections

This section presents the models used to generate our instance benchmarks. The parameters used to generate each benchmark are shown in Table 4.

Table 2: Evaluation on test instances (same size as training) and transfer instances (larger size). We report the geometric mean and standard deviation of the final B&B tree size and the solving time (lower is better for both).

| | Test | | Transfer | |
|---|---|---|---|---|
| Model | Nodes | Time | Nodes | Time |
| SCIP default | $7.3 \pm 39\%$ | $3.3 \pm 10\%$ | $733.9 \pm 26\%$ | $27.4 \pm 7\%$ |
| IL | $52.2 \pm 13\%$ | $2.1 \pm 6\%$ | $805.1 \pm 9\%$ | $14.6 \pm 5\%$ |
| RL (MDP) | $86.7 \pm 16\%$ | $2.2 \pm 6\%$ | $1906.3 \pm 18\%$ | $20.9 \pm 11\%$ |
| RL (tMDP+DFS) | $86.1 \pm 17\%$ | $2.2 \pm 6\%$ | $1804.6 \pm 17\%$ | $20.1 \pm 9\%$ |
| RL (tMDP+ObjLim) | $87.0 \pm 18\%$ | $2.2 \pm 6\%$ | $1841.9 \pm 18\%$ | $20.4 \pm 10\%$ |

Combinatorial auctions

| Model | Nodes | Time | Nodes | Time |
|---|---|---|---|---|
| SCIP default | $10.7 \pm 24\%$ | $5.8 \pm 6\%$ | $61.4 \pm 19\%$ | $12.6 \pm 5\%$ |
| IL | $51.8 \pm 10\%$ | $4.0 \pm 5\%$ | $145.0 \pm 6\%$ | $8.0 \pm 4\%$ |
| RL (MDP) | $196.3 \pm 20\%$ | $5.1 \pm 8\%$ | $853.3 \pm 27\%$ | $14.9 \pm 13\%$ |
| RL (tMDP+DFS) | $190.8 \pm 20\%$ | $5.1 \pm 7\%$ | $816.8 \pm 25\%$ | $14.6 \pm 12\%$ |
| RL (tMDP+ObjLim) | $193.5 \pm 23\%$ | $5.1 \pm 8\%$ | $826.4 \pm 26\%$ | $14.6 \pm 13\%$ |

Set covering

| Model | Nodes | Time | Nodes | Time |
|---|---|---|---|---|
| SCIP default | $19.3 \pm 52\%$ | $13.2 \pm 13\%$ | $2867.1 \pm 35\%$ | $167.4 \pm 23\%$ |
| IL | $35.9 \pm 36\%$ | $8.7 \pm 10\%$ | $1774.8 \pm 38\%$ | $85.7 \pm 22\%$ |
| RL (MDP) | $91.8 \pm 56\%$ | $9.5 \pm 16\%$ | $2768.5 \pm 76\%$ | $85.6 \pm 51\%$ |
| RL (tMDP+DFS) | $89.8 \pm 51\%$ | $9.5 \pm 17\%$ | $2970.0 \pm 76\%$ | $90.6 \pm 51\%$ |
| RL (tMDP+ObjLim) | $85.4 \pm 53\%$ | $9.4 \pm 17\%$ | $2763.6 \pm 74\%$ | $86.1 \pm 47\%$ |

Maximum independent set

| Model | Nodes | Time | Nodes | Time |
|---|---|---|---|---|
| SCIP default | $203.6 \pm 63\%$ | $16.9 \pm 34\%$ | $344.3 \pm 57\%$ | $40.3 \pm 36\%$ |
| IL | $247.5 \pm 39\%$ | $7.2 \pm 26\%$ | $407.8 \pm 37\%$ | $13.6 \pm 24\%$ |
| RL (MDP) | $393.2 \pm 47\%$ | $8.7 \pm 29\%$ | $679.4 \pm 52\%$ | $17.2 \pm 33\%$ |
| RL (tMDP+DFS) | $360.4 \pm 46\%$ | $8.3 \pm 30\%$ | $609.1 \pm 47\%$ | $15.9 \pm 29\%$ |
| RL (tMDP+ObjLim) | $325.4 \pm 41\%$ | $7.9 \pm 26\%$ | $496.0 \pm 48\%$ | $14.5 \pm 28\%$ |

Facility location

| Model | Nodes | Time | Nodes | Time |
|---|---|---|---|---|
| SCIP default | $267.8 \pm 96\%$ | $1.5 \pm 54\%$ | $592.3 \pm 75\%$ | $3.7 \pm 42\%$ |
| IL | $228.0 \pm 95\%$ | $1.8 \pm 66\%$ | $1066.1 \pm 101\%$ | $7.1 \pm 82\%$ |
| RL (MDP) | $143.4 \pm 76\%$ | $1.3 \pm 48\%$ | $518.4 \pm 79\%$ | $4.5 \pm 58\%$ |
| RL (tMDP+DFS) | $135.8 \pm 75\%$ | $1.3 \pm 48\%$ | $495.1 \pm 81\%$ | $4.3 \pm 59\%$ |
| RL (tMDP+ObjLim) | $142.4 \pm 78\%$ | $1.4 \pm 48\%$ | $425.3 \pm 64\%$ | $3.9 \pm 46\%$ |

Multiple knapsack

### A.5.1 Combinatorial auctions

For $m$ items, we are given $n$ bids $\{\mathcal{B}_j\}_{j=1}^n$. Each bid $\mathcal{B}_j$ is a subset of the items with an associated bidding price $p_j$. The associated combinatorial auction problem is of the following form:

$$\text{maximize} \sum_{j=1}^n p_j x_j$$
$$\text{subject to} \sum_{j:i\in\mathcal{B}_j} x_j \leq 1, \quad i = 1, ..., m$$
$$x_j \in \{0, 1\} \; j = 1, ..., n$$

where $x_j$ represents the action of choosing bid $\mathcal{B}_j$.

Table 3: Number of solving runs (instance-seed pairs) out of 200 that hit the 1h time limit.

| Model | C. Auct. | Set Cov. | M.Ind.Set | Fac. Loc. | M. Knap. |
|---|---|---|---|---|---|
| SCIP default | 0 | 0 | 0 | 1 | 0 |
| IL | 0 | 0 | 0 | 0 | 0 |
| RL (MDP) | 0 | 0 | 1 | 0 | 0 |
| RL (tMDP+DFS) | 0 | 0 | 1 | 0 | 0 |
| RL (tMDP+ObjLim) | 0 | 0 | 1 | 0 | 0 |

Test

| Model | C. Auct. | Set Cov. | M.Ind.Set | Fac. Loc. | M. Knap. |
|---|---|---|---|---|---|
| SCIP default | 0 | 0 | 1 | 13 | 0 |
| IL | 0 | 0 | 0 | 0 | 3 |
| RL (MDP) | 0 | 0 | 20 | 1 | 2 |
| RL (tMDP+DFS) | 0 | 0 | 18 | 1 | 0 |
| RL (tMDP+ObjLim) | 0 | 0 | 16 | 1 | 0 |

Transfer

Table 4: Size of the instances used for training and evaluation, for each problem benchmark. We evaluate the final performance on instances of the same size as training (test), and also larger instances (transfer).

| Benchmark | Generation method | Parameters | Train / Test | Transfer |
|---|---|---|---|---|
| Combinatorial auction | Leyton-Brown et al. [24] with arbitrary relationships | Items Bids | 100 500 | 200 1000 |
| Set covering | Balas and Ho [4] | Items Sets | 400 750 | 500 1000 |
| Maximum independent set | Bergman et al. [6] on Erdős-Rény graphs | Nodes Affinity | 500 4 | 1000 4 |
| Facility location | Cornuéjols et al. [8] with unsplittable demand | Customers Facilities | 35 35 | 60 35 |
| Multiple knapsack | Fukunaga [13] | Items Knapsacks | 100 6 | 100 12 |

## A.5.2  Set covering

Given the elements $1, 2, ..., m$, and a collection $\mathcal{S}$ of $n$ sets whose union equals the set of all elements, the set cover problem can be formulated as follows:

$$\text{minimize} \sum_{s \in \mathcal{S}} x_s$$
$$\text{subject to} \sum_{s:e \in s} x_s \geq 1, \quad e = 1, ..., m$$
$$x_s \in \{0, 1\} \ \forall s \in \mathcal{S}$$

## A.5.3  Maximum independent set

Given a graph $G$ the maximum independent set problem consists in finding a subset of nodes of maximum cardinality such that no two nodes in that subset are connected. We use the clique formulation from [6]. Given a collection $\mathcal{C} \subseteq 2^V$ of cliques whose union covers all the edges of the

graph $G$, the clique cover formulation is

$$\text{maximize} \sum_{v \in V} x_v$$

$$\text{subject to} \sum_{v \in C} x_v \leq 1, \quad \forall C \in \mathcal{C}$$

$$x_v \in \{0, 1\} \quad \forall v \in V$$

### A.5.4 Capacitated facility location with unsplittable demand

Given a number $n$ of clients with demands $\{d_j\}_{j=1}^n$, and a number $m$ of facilities with fixed operating costs $\{f_i\}_{i=1}^m$ and capacities $\{s_i\}_{i=1}^m$, let $c_{ij}/d_j$ be the unit transportation cost between facility $i$ and client $j$, and let $p_{ij}/d_j$ be the unit profit for facility $i$ supplying client $j$. We try to solve the following problem

$$\text{minimize} \sum_{i=1}^m \sum_{j=1}^n c_{ij} x_{ij} + \sum_{i=1}^m f_i y_i$$

$$\text{subject to} \sum_{j=1}^n d_j x_{ij} \leq s_i y_i, \quad i = 1, ..., m$$

$$\sum_{i=1}^m x_{ij} \geq 1, \quad j = 1, ..., n$$

$$x_{ij} \in \{0, 1\} \quad \forall i, j$$

$$y_i \in \{0, 1\} \quad \forall i$$

where each variable $x_{ij}$ represents the decision of facility $i$ supplying client $j$'s demand, and each variable $y_i$ representing the decision of opening facility $i$ for operation.

### A.5.5 Multiple knapsack

Given $n$ items with respective prices $\{p_j\}_{j=1}^n$ and weights $\{w_j\}_{j=1}^n$, and $m$ knapsacks with capacities $\{c_i\}_{i=1}^m$, the multiple knapsack problem consists in placing a number of items in each of the knapsacks such that the price of the selected items is maximized, while the capacity of the knapsacks is not exceeded by the total weight of the items therein. Formally:

$$\text{maximize} \sum_{i=1}^m \sum_{j=1}^n p_j x_{ij}$$

$$\text{subject to} \sum_{j=1}^n w_j x_{ij} \leq c_i, \quad i = 1, ..., m$$

$$\sum_{i=1}^m x_{ij} \leq 1, \quad j = 1, ..., n$$

$$x_{ij} \in \{0, 1\} \quad \forall i, j$$

where each variable $x_{ij}$ represents the decision of placing item $j$ in knapsack $i$.