# OpenReview forum: "Learning to Branch with Tree MDPs"
_NeurIPS.cc/2022/Conference — NeurIPS 2022 Accept_

### Official Review · Reviewer_NiZ2 · 2022-07-10

**Rating:** 8
**Confidence:** 4
**Soundness:** 4 excellent
**Presentation:** 3 good
**Contribution:** 3 good

**Summary:**

The main contribution of this paper is a new type of augmented MDP formulation called a “tree MDP”, which is motivated by the “learning to branch” problem in mixed-integer linear programming (MILP). The intuition is to convert the default MDP formulation, which we can think of to be defined over subtrees, into a simpler formulation that is defined over nodes instead. The authors then show computational experiments on a set of benchmarks, showing that the new MDP formulation is more amenable for RL.

**Questions:**

Questions:
[Side-by-side comparison] It would be nice to have a side-by-side comparison between the temporal MDP formulation and the tree-MDP formulation, where the states, actions, reward, and definition of episode are compared.
[Requirement of optimal solutions] In Section 3.2, it is mentioned that one of the drawbacks of the standard approach is that “...require to run episodes to completion, that is, to solve MILP instances to optimality, which leads to very long episodes even for moderately hard instances.” However, based on my understanding, the ObjLim version also requires this so it would be nice to properly contextualize some of this (e.g., “with a certain assumption, we can get rid of this requirement”). For the side-by-side comparison mentioned in the point above, it would also be good to mention which assumptions are needed.
[Deterministic MDP formulation] As I was reading through Section 4.1, I was a little thrown off by what is meant by p_ch^- and p_ch^+ since I expected them to be deterministic. It is later specified that the MDP is indeed deterministic (line 215). I wonder if it would be clearer to define a tree MDP only for the deterministic case, rather than in full generality. Is there any point that we need a stochastic tree MDP?
[Other applications of tree MDP] Related to the above, can the authors point to other applications where a tree MDP might be useful? If so, this would be a good reason to present the general setup of a tree MDP.
[Exponentially sized episodes and more difficult policy gradient computation] The paper emphasizes that the states and rewards for a tree MDP are localized to the specific nodes. It would be good to also emphasize that the trade-off here is that each episode is now a tree and therefore can blow up in size -- this trade-off now makes the policy gradient computation more difficult, since it requires a sum over all nodes. If I misdiagnosed the drawback, please correct me, but my concern here is that a cursory read might suggest to a reader that all aspects are simplified compared to before.
[Temporal RL algorithms] In a policy gradient setting, I can see that a tree MDP can be used because the entire episode is run and then (s, a, G) are extracted. Could tree MDPs be used with algorithms like Q-learning that make stronger use of the temporal MDP setup? It is not clear to me that they could because in order to compute an immediate reward, one would need to wait for both child states to be evaluated. Some clarity / discussion on this would be nice!
[Capitalized states and actions] In line page 5, line 182, the states and actions are capitalized -- is this a typo or does it have some significance?
[Figure 1] In Figure 1, the dotted lines are not explained (although I realized that they mean the node was pruned). It would be good to clarify.
[GNN] What exactly is the state fed to the policy for each of the different algorithms? I was confused by the statement that “same features and architecture” for the GNN are used, but I thought that the states for the MDP / tree MDP approaches are quite different.
[Computational results] What does 1804.6 +/- 17% mean? Is the 17% defined to be 1804.6 +/ 0.17*1804.6? Why do we use geometric mean here? Would the results be different if sample mean were used?

**Limitations:**

They have to some extent, but I have also asked for a few more points in the Questions section above.

**Strengths And Weaknesses:**

Strengths: I found the tree MDP idea used in the paper to be novel, creative, and directly targets the motivating problem without extraneous features and believe that this is a valuable contribution to the literature on applying RL to MILP.

Weaknesses: The ideas in the paper took me some time to properly digest. I believe (nearly) all of the information needed for the reader to digest is there, but think that the paper could make this process easier on the reader. Please see the questions below for specific queries.

---

> ### Author Response · Authors · 2022-08-02
> **Response to reviewer NiZ2 (2/2)**
>
> - **GNN**
> Thanks for pointing out this confusion. We do indeed train branching policies using the same features and architecture in every approach, MDP or tMDP. The reason is as follows. In order to formulate branching as an MDP, state $s_t$ is the partial B\&B tree up until step $t$. This ensures that the Markov property holds. In Section 4.2 we state that, using $s_i=(\\textit{MILP}_i, \\textit{GUB}_i)$ (this is, a subset of the MDP state) and under assumption 4.2, we can formulate branching as a tree MDP.  In summary, the states for the MDP / tree MDP approaches are indeed quite different (the second being a subset of the first).
> However, when it comes to implementation, the MDP state is impractical to handle and it is common practice to use only a subset of it to train branching policies (see Gasse et al. (NeurIPS'19) or Khalil et al. (AAAI'16)), which effectively turns the MDP into a POMDP. This is an additional argument in favor of tMDPs, where the state is simpler, and easier to use. Here we use the same features and architecture (Gasse et al., NeurIPS'19), which provides an efficient and flexible representation of the local MILP, as well as the global upper bound. We have added a clarification in the experiments section.
>
> - **Computational results**
> Here 1804.6 is the geometric mean over all samples (all instance-seed combinations), while 17\% is the standard deviation (in percentage over mean) per instance. We follow the reporting metrics used in (Gasse et al., NeurIPS'19). The geometric mean is the metric of choice for comparing tree sizes resulting from MILPs, as established by Achterberg and Wunderling (2013). This prevents outliers from skewing the comparison excessively.

---

> > ### Comment · Reviewer_NiZ2 · 2022-08-10
> > **Thank you**
> >
> > Thank you for the detailed responses! After my initial review and reading the responses + changes to the paper, I believe that the paper makes a valuable contribution and would like to increase my score.

---

> ### Author Response · Authors · 2022-08-02
> **Response to reviewer NiZ2 (1/2)**
>
> We thank the reviewer for their valuable time and feedback. We have made modifications to the paper that we hope improve clarity and presentation. Below we provide answers to the questions raised in the review. We would be happy to provide further clarifications if suitable.
> - **Side-by-side comparison**
> We completely agree. We have added a side-by-side comparison of the components that define a temporal MDP and a tree MDP. Due to the very limited space in the main text, this aid can be found in the supplementary material (Section A.1).
>
> -  **Requirement of optimal solutions**
> If one chooses final tree size as a metric to be used as a return, solving each MILP to optimality is a requirement independently of the chosen formulation (MDP or tMDP). In Section 3.2, we mention this as a fundamental issue of learning to branch with RL, which contributes to (1) computationally expensive sample generation, and (2) a particularly difficult credit assignment task. Our tMDP formulation addresses only the latter point. It is true, however, that imposing ObjLim, as we propose, effectively results in shorter episodes. This was indeed unclear from the text and we have therefore added a clarification at the end of Section 3.2. We thank you for pointing this out.
>
> - **Deterministic MDP formulation**
>  Indeed, we define a stochastic tMDP for the sake of generality. The goal is to stress the wider applicability of our formulation. It also provides a more natural extension from MDPs, which are usually defined to have stochastic transitions. It is true however that we can improve clarity by explicitly indicating that we deal with a deterministic tMDP from the start. We added a footnote in Section 4.1 to clarify this point.
>
> - **Other applications of tree MDP**
> This is an excellent point, which we did not really address in the paper. Since another reviewer asked essentially the same question, we answered as a general comment, which we refer you to. We also decided to expand on this topic in the paper, by adding some suggestions of further applications in the conclusion (Section 6). We hope you find our discussion addresses your concerns.
>
> - **Exponentially sized episodes and more difficult policy gradient computation**
> Thank you for pointing this aspect out. We would like to highlight the fact that an episode of a tree MDP is a tree does not increase the difficulty in terms of the size. The episode length is equal to the B\&B tree size in both formulations. However, it is true that computing the policy gradients, and in particular the returns (sum of rewards) in tree MDPs requires storing and using the tree structure of the episode, while in temporal MDPs the episode is simply an ordered list. Still, the computation of the returns can be done efficiently using a bottom-up traversal of the tree, which has complexity $O(n)$, same as for MDPs. Of course the recursive implementation for tree MDPs is a little bit more involved than for temporal MDPs. We added a clarification in the form of a footnote to Algorithm 1.
>
> - **Temporal RL algorithms**
> Q-learning is indeed compatible with tree MDPs. With an MDP formulation, one must store $(s_t, a_t, r_t, s_{t+1})$ tuples.
> In the case of tree MDPs one would instead need $(s_i, a_i, r_i, s_{ch_i^-}, s_{ch_i^+})$ . The Q-learning update rule simply becomes
> $$
> Q_{new}(s_i, a_i) = Q(s_i, a_i) + \\alpha (r_i + \\max_a Q(s_{ch_i^-}, a) + \\max_a Q(s_{ch_i^+}, a) - Q(s_i, a_i))
> \\text{.}
> $$
> Computing the immediate rewards $r_i$ does not necessarily require to wait for the children nodes to be processed. For example, when minimizing the tree all immediate rewards are -1. When minimizing solving time, the immediate reward is obtained simply by solving the node's local LP, under mild assumptions (see Section 4.2.2 in the paper). Finally, we would like to point out that Etheve et al. (2020) apply a Q-learning algorithm for learning to branch, in a framework that turns out to be a tree MDP.
>
> - **Capitalized states and actions**
> Thank you for the comment. We have added a clarification at the beginning of Section 2.2 regarding our notation, which we had not stated in the previous version of the paper.
>
> - **Figure 1**
> After some consideration we decided to make all lines in Figure 1 solid, to make the figure simpler and improve readability. Thank you for pointing this out.

---

### Official Review · Reviewer_hf9x · 2022-07-11

**Rating:** 6
**Confidence:** 3
**Soundness:** 3 good
**Presentation:** 3 good
**Contribution:** 3 good

**Summary:**

The paper proposes a tree MDP formulation for learning to branch in MILP branch-and-bound algorithms. A policy gradient algorithm for solving MILP tree MDPs is proposed. Conditions for the correctness of the algorithm are derived. The paper compares the tree MDP formulation with the vanilla RL formulation as well as a strong baseline (default SCIP) on 5 benchmark problem sets. The main empirical result shows that the tree MDP formulation outperforms the vanilla RL formulation but does not outperform SCIP in 4 out of 5 problems. Similar results are observed for the transfer learning experiments.

--------

UPDATE: I thank the authors for their detailed responses. After reading the other reviews and the responses, I'm a bit more positively inclined towards the paper although the computational performance remains a bit of an issue for me. I've upgraded my score to a Weak Accept.

**Questions:**

1. How novel is the tree MDP formulation?
  - What, if any, relationship, does tMDP have to a recursively-optimal hierarchical RL algorithm like MAXQ [Dietterich99])?

2. Does the tree MDP formulation have other applications besides B&B trees for MILPs?

3. Why does tMDP outperform SCIP default on Multiple Knapsack? Is it possible to characterize how the branching rule learned in tMDP is overcoming the poor relaxation (assuming that's the root cause)? Can the presence / absence of the poor relaxation be verified experimentally?

**Limitations:**

There is a discussion on the limitations and social impact in the checklist (but not the main paper). The authors could consider moving it to the main paper.

**Strengths And Weaknesses:**

Strengths
  + There appear to be a number of novel algorithmic contributions (tree MDP, using tree MDPs to encode B&B trees for MILPs, policy gradient algorithm for tree MDPs).
  + The proposed approach seems well-motivated, intuitively clear with correctness guarantees for ObjLim and and DFS (although I didn't check these carefully). Combined with the novelty, this makes the paper interesting in itself.
  + The paper is tackling a problem of significant practical importance (B&B, MILPs) and improvements here would have a large impact.
  + The paper is clearly written and the proposed ideas and experiments are easy to follow.

Weaknesses
  - The empirical results don't show improved performance over the baseline in 4 out of 5 cases. In the case where it does well (Multiple Knapsack), there is no deeper analysis. Overall, the experimental section provides limited insight into the empirical characteristics of tMDP compared to SCIP.
  - It's unclear if the tree MDP formulation can be used in other applications. The paper does not place the tree MDP formulation into the broader (and large) body of work on MDPs. As a result, it becomes difficult to assess the impact of the algorithmic contributions. (This is reflected in my score for presentation and contribution.)
  - The baseline (SCIP branching rule) is not described and the description strong branching is very brief. Coupled with the above, it becomes difficult to place the proposed ideas into the existing body of knowledge.

---

> ### Author Response · Authors · 2022-08-02
> **Response to reviewer hf9x (2/2)**
>
> > How novel is the tree MDP formulation? What, if any, relationship,
> does tMDP have to a recursively-optimal hierarchical RL algorithm like MAXQ [Di-
> etterich99])?
>
> Thank you for raising this interesting point. In MAXQ, the authors propose to decompose
> the main task into a set of simpler tasks which can be solved recursively, independently of the
> parent task (e.g., pick up and deliver a package from A to B decomposes to: move to A, pick up,
> move to B, drop). Both approaches have similarities, in the sense that they exploit a hierarchical
> decomposition of the task at hand in order to simplify the credit assignment problem in RL.
> However, the two methods also differ on several points. 1) in MAXQ, the hierarchical sub-task
> decomposition must be given a priori by the user for each new task, and is set in stone with a
> limited depth, while in tree MDPs the decomposition holds by construction, and can be applied
> recursively for virtually infinite depths; 2) in MAXQ, the subtasks are different (different reward
> and optimal policy), while in tree MDPs the reward remains the same; 3) in MAXQ, each sub-task
> necessarily results in a series of consecutively processed states (e.g., AAABBBCC), while in tree
> MDPs the temporal processing order of states can vary, and switches between different sub-trees
> are allowed (e.g., AACBBAC); and 4) in MAXQ, the resulting process is made Markovian by
> including the subtask stack K to the state S, while in tree MDPs the state S is sufficient to have
> the Markovian property. We propose to highlight better those differences in the final version of
> the manuscript, in a new Section 4.4 ”Relationship with hierarchical RL approaches”.
>
> > Does the tree MDP formulation have other applications besides B&B trees for
> MILPs?
>
> We refer you to the general comment on this topic, as well as our answer to your second
> weakness.
>
> > Why does tMDP outperform SCIP default on Multiple Knapsack? Is it possible
> to characterize how the branching rule learned in tMDP is overcoming the poor
> relaxation (assuming that’s the root cause)? Can the presence / absence of the poor
> relaxation be verified experimentally?
>
> SCIP’s default rule chooses a variable based on multiple criteria, but assigns great importance
> to pseudocost information (see newly added seection A.4). Pseudocosts measure the change in
> objective value of the LP relaxation that we incur in when branching. In the case of multiple
> knapsack we observed that fixing a variable very often results in no change in optimal value of
> the LP relaxation. More specifically, we tested a subset of the instances to find that pseudocosts
> were exactly zero in 99.8% of the cases. Consequently, the brancher must select a variable using a
> criterion that is very often not discriminative. Our policy is based on a more diverse set of problem
> data, and is able to learn to use it effectively through self-learning, while SCIP default and the
> imitation learning approach fail. This was indeed not clear enough from the text and we thank
> you for pointing it out. We hope that the added clarification in the experiment section, together
> with the new section A.4, address this point sufficiently.

---

> ### Author Response · Authors · 2022-08-02
> **Response to reviewer hf9x (1/2)**
>
> We thank the reviewer for their time, and their excellent insights regarding the paper. We will address each concern in order.
>
> > The empirical results don’t show improved performance over the baseline in 4
> out of 5 cases. In the case where it does well (Multiple Knapsack), there is no deeper
> analysis. Overall, the experimental section provides limited insight into the empirical
> characteristics of tMDP compared to SCIP.
>
> We understand this criticism. However, we would like to point out that our baseline in this
> paper is vanilla RL with temporal MDP, not SCIP. Our main contribution is providing a strategy
> to make RL for branching more efficient. Our experiments demonstrate that the tree MDP formulation consistently improves the performance of a standard RL algorithm for learning to branch, on
> all 5 benchmarks. But our contribution does not solve the learning problem entirely, and obtaining
> state-of-the-art solver performance via RL requires to tackle additional challenges, such as the high
> computational cost for collecting training episodes, the parameterization of the branching policy,
> or simply the engineering challenges that must be faced for scaling. We acknowledge this in the
> paper and we point out that more research and efforts are needed in order to bridge the gap and
> fully unlock potential of RL for branching. Nevertheless, we also firmly believe that this paper
> opens a promising path in this direction, which will be of value to the community even without
> state-of-the-art performance against SCIP.
>
> > It’s unclear if the tree MDP formulation can be used in other applications. The
> paper does not place the tree MDP formulation into the broader (and large) body of
> work on MDPs. As a result, it becomes difficult to assess the impact of the algorithmic
> contributions. (This is reflected in my score for presentation and contribution.)
>
> This is a good point: our paper was missing a discussion regarding the larger context in which
> this construction (the tree MDP) belongs, and potential applications beyond controlling decision
> tasks in branch-and-bound MILP solvers. Since another reviewer asked a very similar question
> regarding other applications, we preferred to provide a joint answer as a general comment, which
> we refer you to. We also expanded on this topic directly in the article, by adding a discussion of
> further applications in the conclusion. We hope that this addresses your concern. Regarding the
> connections with other MDP variants, we could not find any prior construction that encapsulates
> this tree MDP concept: as far as we were able to find, our construction is novel. The closest
> connection is perhaps hierarchical reinforcement learning, as you suggest in your first question:
> nonetheless, there are differences, which we address in our answer below.
>
> > The baseline (SCIP branching rule) is not described and the description strong
> branching is very brief. Coupled with the above, it becomes difficult to place the
> proposed ideas into the existing body of knowledge.
>
> The default branching rule of SCIP (which is not our baseline, see our answer above) is
> named reliability pseudocost branching (sometimes referred to as hybrid branching), and consists
> in a complex combination of heuristic rules based on so-called conflict scores, inference scores,
> cutoff scores, pseudo-cost scores and partial strong branching scores of the branching variables.
> The paper being more targeted at the machine learning community, we found it nonessential to
> describe this branching rule in great detail, and we focused on placing our work into the learning
> to branch literature. However, we agree that some readers might find it useful to have a more
> detailed description of SCIP’s branching rule, so we have added a section in the supplementary
> material (A.4) with a brief discussion of its characteristics. Thank you for pointing this out.

---

### Official Review · Reviewer_Je2V · 2022-07-12

**Rating:** 6
**Confidence:** 3
**Soundness:** 3 good
**Presentation:** 3 good
**Contribution:** 2 fair

**Summary:**

This paper proposes a new Tree MDP theory and formalizes the branching problem as a Tree MDP problem.
Under this new formulation, the resulting algorithm benefits sample efficiency and credit assignment.

**Questions:**


Questions:
My major concern is how could the method be generalized to the test environment? In other words, does there exists an optimal policy that can map the MIP problem into an action?

Although the authors have discussed the limitations in Section 4.4, I think this is the fundamental point about the method, as it is the key to the soundness of the method. I think the paper should discuss more on that.

Some related questions are as follows:

1. What's the architecture of policy \pi_\theta in your implementation?
2. In line 190, the state is embedded as MILP_i, GUB_i. How to embed a MILP_i? This is a critical problem for generalizing the policy for testing.

Some points are not clear to me:

1. It is not clear to me why a state (MILP_i, GUB_i) can carry enough information to unroll a vanilla B&B algorithm.
2. For a B&B tree in Figure 1, if h is the current state, then it expands from h. After several iterations, is that possible that the policy will still expand d or c or g?

**Limitations:**



**Strengths And Weaknesses:**


This paper proposes a novel Tree MDP theory, which is well fitted for the branching problem.
The new tree MDP formulation has an obvious advantage over the existing MDP formulation.

However, the paper has some basic theoretical flaws.

My major concern is how could the method be generalized to the test environment? In other words, does there exists an optimal policy that can map the MIP problem into an action?

Assumption 4.2 is strong to me. The paper has offered two conditions in Proposition 4.4 and 4.5.

However, in P 4.4, if the optimal value is provided, the problem is reduced to another easier problem with equality constraint (is this true?).
In P 4.5, the algorithms can be limited to a very limited scope.

---

> ### Author Response · Authors · 2022-08-02
> **Response to reviewer Je2V**
>
> We thank the reviewer for their valuable time and feedback.
> The reviewer is mainly concerned about generalization to the test environment. This is an excellent point, as indeed the test environment will not in general have the tree Markov property. We believe that the reasons why the gains seem to generalize to the test environment (as shown in our experiments) are as follows. First, the node selection rules used in practice by solvers (such as SCIP's default rule, "best estimate'') exhibit diving behavior that resembles DFS. In addition, the gains brought by primal heuristics are often the most significant early on during solving, so that past a certain point there is little interaction between the branches of the B\&B tree. Thus, one would expect in general that the tree Markov property should approximately hold for one or the other reason, or both, and this can help provide a theoretical explanation why the improvements brought by switching to a tree RL algorithm generalize to the test environment.
>
> We have also made modifications to the paper that we hope improve clarity and presentation. Below we provide answers to the questions raised in the review. We would be happy to provide further clarifications if suitable.
>
> > What's the architecture of policy $\pi_\theta$ in your implementation?}
>
> The architecture is identical to the one used in (Gasse et al., NeurIPS'19). We refrained from making architectural changes, as architecture optimization is outside the scope of this paper.
>
> >  In line 190, the state is embedded as $MILP_i$, $GUB_i$. How to embed a $MILP_i$? This is a critical problem for generalizing the policy for testing.}
>
> Following (Gasse et al., NeurIPS'19), we represent the local MILP as a bipartite graph $G=(V\cup C, E)$. Each problem variable is represented by a node $v\in V$. Likewise, each constraint has an associated node $c\in C$. An edge is drawn between a node $v\in V$ and a node $c\in C$ if the corresponding variable has a non-zero coefficient in the corresponding constraint. Each node has an associated feature vector that contains problem information such as bounds, type and status. This representation has been shown to provide a very efficient and flexible representation of a MILP. For more details we refer to (Gasse et al., NeurIPS'19).
>
> > It is not clear to me why a state $(MILP_i, GUB_i)$ can carry enough information to unroll a vanilla B\&B algorithm.}
>
> To implement a B\&B algorithm, one simply needs to store in memory the current global upper bound ($GUB$), as well as the current partitioning of the original MILP, which consists here in the B\&B tree structure and the sub-MILP $(MILP_i)$ associated to each of its nodes. Then, running B\&B means selecting an open node (LP relaxation below the GUB), splitting this node into two children nodes with their own sub-MILPs, solving their LP relaxations and updating the GUB if needed. In practice, much more information is computed and stored by a B\&B solver for efficiency. However the only information that is essential to run B\&B is the tree structure, the sub-MILPs and the GUB (which can be extracted as the minimum of all $GUB_i$).
>
> > For a B\&B tree in Figure 1, if h is the current state, then it expands from h. After several iterations, is that possible that the policy will still expand d or c or g?}
>
> The only restriction is that a node, say $h$, can only be processed after all its parent nodes have been processed, hence $a$, $c$ and $f$ here. So after node $h$ is processed, it is possible that nodes $d$ and $g$ will be processed if they haven't been already. Note that we meant the B\&B tree in Figure~1 to be the final one, that is, nodes $d$, $e$, $h$, $i$, $g$ are leaf nodes which will not yield any child node after they are processed (hence node $h$ will not be expanded).

---

### Author Response · Authors · 2022-08-02
**About the applicability of tree MDP to other domains**

Two reviewers sensibly wondered whether the tree MDP framework we propose in this work could be applicable to other problems, besides modelling decision tasks in B\&B based MILP solvers. We answer as a general comment here rather than duplicating the answer.

A tree MDP model is applicable whenever one has a control problem, where the problem subdivides recursively into tasks controlled by the same policy. At minimum, this is the case for divide-and-conquer algorithms, a category to which branch-and-bound belongs, so we would expect it to be a good fit for modelling any problem where the objective is to control some aspect of this larger class of algorithms.

For example, one could imagine a robotics problem where a rover must explore interconnected rooms to perform some action, and must explore as efficiently as possible. The control task of learning to decide which rooms to explore could be modelled as a tree MDP. Another, more abstract example could be choosing the pivot elements in quicksort, a divide-and-conquer sorting algorithm. At each time step of this algorithm, the pivot is used to partition the set of elements into two, and careful selection of the pivot can have a dramatic impact on the algorithm efficiency. This is another problem that would be a good fit for a tree MDP.

These are only two examples, but we believe that the framework could be useful for many more. In each case, using a regular MDP framework would be possible, but switching to a tree MDP formulation and its associated RL algorithms could lead to better credit assignment and hence, to better sample efficiency.

We have added a short discussion of these examples in Section 6.

---

### Meta-Review · Area_Chair_NCFv · 2022-08-27

**Recommendation:** Accept
**Confidence:** Certain

**Metareview:**

The paper studies the MILP problem by providing a tree MDP framework for a more suitable formulation of the branching problem.
The reviewers believe that this approach is relevant and novel. While in the first round of the review, the reviewers had identified a number of concerns such as the applicability of the tree MDP, concerns about the comparison of baselines, and presentation clarities. The authors have addressed these issues in a satisfactory way in the rebuttal phase. All the reviewers unanimously agree to accept the paper.

**Award:**

No

---

### Decision · Program_Chairs · 2022-09-14

Accept